# Production of Butyric Acid from Hydrolysate of Rice Husk Treated by Alkali and Enzymes in Immobilized Fermentation by *Clostridium tyrobutyricum* Ct∆*pta*

**Yueying Lin** [1]**, Wanjing Sun** [1]**, Geng Wang** [1]**, Haohan Chen** [1]**, Xun Pei** [1]**, Yuyue Jin** [1]**, Shang-Tian Yang** [2]
**and Minqi Wang** [1,*]

[1]   College of Animal Science, Zhejiang University, Hangzhou 310058, China
[2]   Department of Chemical and Biomolecular Engineering, The Ohio State University, Columbus, OH 43210, USA
[*]   Correspondence: wangmq@zju.edu.cn; Tel.: +86-571-88982112; Fax: +86-571-88982650

**Abstract:** Rice husk, as a cellulose-rich by-product in agriculture, has been considered as a low-cost substrate for the production of bioethanol and chemicals. In this study, rice husk was pretreated with an alkali, followed by cellulose and *β*-glucosidase hydrolysis optimized by an orthogonal experiment and response surface methodology (RSM), respectively. Under the optimal treatment conditions, a hydrolysate containing a high reducing sugar yield (77.85%) was obtained from the rice husk. Then, the hydrolysate was used as a carbon substrate for butyric acid production through *Clostridium tyrobutyricum* ∆*pta* fermentation. Compared to free-cell fermentation, higher concentrations of butyric acid (50.01 g/L vs. 40.8 g/L and 49.03 g/L vs. 27.49 g/L) were observed in immobilized-cell fermentation for the carbon source of glucose and hydrolysate, respectively. A final butyric acid concentration of 16.91 g/L, a yield of 0.31 g/g, and an overall productivity of 0.35 g/L/h from rice husk hydrolysate were obtained in the repeated-fed-batch mode. Taken together, rice husk hydrolysate can be effectively utilized for the bioproduction of butyrate with immobilized-cell fermentation.

**Keywords:** rice husk; *Clostridium tyrobutyricum*; butyric acid; fibrous bed bioreactor

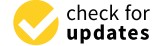



## 1. Introduction

Butyric acid is a four-carbon volatile fatty acid that has been extensively applied in the food, pharmaceutical, and chemical industries. Currently, butyric acid can be produced through the oxidation of butyraldehyde and microbial fermentation [1]. However, the limited supply of petroleum and consumers' opposition to food additives containing synthetic ingredients have driven industrial attention toward fermentation-derived butyric acid.

The global biomass resources are rich, of which the energy stored in the form of lignocellulose can account for more than 60% [2]. Lignocellulose biomass, such as agricultural residues, is cheap and largely available. Since the substrate cost accounts for a large proportion of butyric acid production, renewable feedstocks such as saccharina japonica [3], coffee ground [4], rape straw [5], and sugarcane bagasse [6] have been used for its production. However, lignocellulose biomass is difficult to ferment directly due to its complicated structure and tight connection [7]. In general, chemical pretreatment and enzymatic hydrolysis are beneficial to remove lignin or disrupt bonds in the lignocellulose [8]. Therefore, the appropriate pretreatment and fermentation process could improve the utilization of lignocellulose.

Anaerobic microorganisms that produce butyric acid as the main end-product are often *Clostridium species* [9]. Among them, *C. tyrobutyricum* has been proven to produce butyric acid from fermentable sugars with a higher efficiency [10]. However, the conventional

butyric acid fermentation process displayed low yield and productivity. Subsequently, integrational mutagenesis and cell immobilization became promising strategies to optimize its production, especially integrational mutagenesis, which could shift metabolic flux towards the target product [11]. An early attempt to inactivate the *ack* gene, the key enzyme in the acetic acid metabolic pathway, resulted in about a 30 % increase in butyrate production [12]. In addition, many previous works pointed out that the cell immobilization technology was used for the enhancement of the production of biohydrogen [13], glutaric acid [14], and lactic acid [15].

The objective of this study was to evaluate the feasibility of butyric acid production by fermentation fed with rice husk. First, the alkali pretreatment and enzymatic hydrolysis of rice husk were optimized to improve its total reducing sugar (TRS) yield. Then, fermentation performances with different carbon sources in both the free-cell and immobilized-cell modes were investigated. Finally, rice husk hydrolysate was utilized in the repeated-batch mode by *C. tyrobutyricum Δpta* immobilized in a fibrous bed bioreactor (FBB) to evaluate the process stability and long-term performance.

## 2. Materials and Methods

### 2.1. Materials

Rice husk was dried in an oven until the water content was less than 5%, ground to a fine powder, and passed through a 40-mesh (0.425 mm) screen. *β*-glucosidase (Ruiyang Biotechnology Co., Ltd., Shanghai, China) with an activity of 26 U/g and cellulase (Habio Biotechnology Co., Ltd., Sichuan, China) with the activity of 90 U/g were used in this experiment. Other chemicals were purchased from Sinopharm Chemical Reagent Co., Ltd., Shanghai, China, which were analytical-grade.

### 2.2. Single-Factor and Orthogonal Test of the Alkali Pretreatment for TRS Production

For the alkali pretreatment with NaOH, the rice husk was treated with different reaction times (1, 2, 3, and 4 h), alkali concentrations (2, 3, 4, and 5%), and temperatures (100, 110, 120, and 130 °C). Then, the residues after the alkali pretreatment were washed by ddH$_2$O, neutralized with 3 M dilute sulfuric acid, and dried in an oven at 65 °C. For each gram of the above residue, 10 U of cellulose, 10 U of *β*-glucosidase, and 20 mL of 50 mM sodium citrate buffer were needed for the enzymatic hydrolysis. After 36 h of reaction in the thermostatic gas bath rocking bed (50 °C), the supernatant was collected after centrifuging (3000 rpm) for 5 min. The TRS content in the supernatant was detected by the DNS method.

Based on the results of the above single-factor experiment, an orthogonal design of an L9 ($3^3$) (shown in Table 1) array was designed to further optimize the alkali pretreatment. The analysis of variance was performed with SPSS 22.0 (SPSS Inc., Chicago, IL, USA).

**Table 1.** Variables of alkali pretreatment in the orthogonal design.

| | Levels | | |
|---|---|---|---|
| **Independent Variables** | **Low** | **Medium** | **High** |
| A = Time (h) | 2 | 3 | 4 |
| B = Alkaline concentration (wt%) | 2 | 3 | 4 |
| C = Temperature (°C) | 110 | 120 | 130 |

### 2.3. Optimization of Enzymatic Hydrolysis of Rice Husk for TRS Production

The residues obtained from the alkali pretreatment under the optimized condition were used for the following enzymatic hydrolysis. At first, the effects of different parameters, including the reaction time, ratio of cellulase to *β*-glucosidase, total amounts of the cellulase and *β*-glucosidase, and temperature, were evaluated on TRS yield, and the determination of the TRS in the supernatant was identical to Section 2.2. Referring to the TRS yield in the above single-factor experiment, the enzyme amount (A), temperature (B), and reaction

time (C) were selected for a further optimization process by response surface methodology (RSM). The Box–Behnken experimental design was adopted for RSM, and the details are shown in Table 2. The response surface analysis was conducted using Design Expert (Version 8.0.6, Stat-Ease Inc., Minneapolis, MN, USA).

**Table 2.** Variables of enzymatic hydrolysis in the Box–Behnken design.

| Independent Variables | Levels | | |
| --- | --- | --- | --- |
| | Low | Medium | High |
| A = Enzyme amount (U) | 40 | 50 | 60 |
| B = Temperature (°C) | 50 | 60 | 70 |
| C = Time (h) | 36 | 48 | 60 |
| TRS yield (%) | maximize | | |

### 2.4. Strain Culture

A *pta* knockout mutant of *C. tyrobutyricum* ATCC 25755 (*C. tyrobutyricum* Δ*pta*) used in this work was previously constructed according to the method in [16]. The strain was cultured in a synthetic medium that consisted of 60 g/L glucose, 2 g/L yeast extract, 4 g/L tryptone, 2 g/L $(NH_4)_2SO_4$, 1 g/L $K_2HPO_4$, 0.1 g/L $MgSO_4 \cdot 7H_2O$, 0.15 g/L $FeSO_4 \cdot 7H_2O$, 0.015 g/L $CaCl_2 \cdot 2H_2O$, 0.01 g/L $MnSO_4 \cdot H_2O$, 0.02 g/L $CoCl_2 \cdot 6H_2O$, and 0.002 g/L $ZnSO_4 \cdot 7H_2O$. The cultivation conditions were kept at 37 °C, and the initial pH was 6.8 anaerobically.

### 2.5. Free-Cell Fermentations

Free-cell fermentations fed with glucose and concentrated rice husk hydrolysate were conducted in a 5 L stirred-tank fermentor (BIOTECH-5BG, Baoxing Bioengineering Equipment Co., Ltd., Shanghai, China). Prior to fermentation, the content of the pretreated rice husk was filtered to separate the liquid fractions and solid fractions with the centrifuge. The liquid fractions were concentrated using vacuum concentration at 60 °C and sterilized by ultraviolet light. The sterilized media were flushed with $N_2$ for 30 min to remove oxygen then inoculated with 100 mL of active seed culture. The fermentation kinetics were studied in the fermentor at 37 °C (130 rpm), and the pH value was kept between 4.5 and 6 by manually adding 6 M $NH_3 \cdot H_2O$. The carbon source consumption and cell growth were monitored with liquid samples taken periodically throughout the fermentation.

### 2.6. Immobilized-Cell Fermentations in FBB

Immobilized-cell fermentations fed with the two abovementioned substrates were carried out by connecting the FBB to the fermentor. The experimental procedure was previously described in [17]. The FBB construction contained a glass column, stainless-steel mesh, and a piece of spirally cotton towel (230 × 300 mm, 5 mm in thickness, and 95% porosity) (Figure 1). Before use, the fermentor was autoclaved twice and connected to the FBB by tubes. A 100 mL volume of cell suspension was incubated in the fermentor and immobilized in the cotton towel by recirculating the medium between the fermentor and the FBB at the rate of 25 mL/min. The fermentation was carried out with 2 L of the synthetic medium at 37 °C with the pH controlled by adding 6 M $NH_3 \cdot H_2O$. The broth was quickly replaced with fresh medium for further fermentation kinetics studies when the cell density ceased to drop. Fermentations with different carbon sources (glucose and concentrated rice husk hydrolysate) under free-cell and immobilized-cell modes were first studied, and the fermentation process was continued until *C. tyrobutyricum* Δ*pta* stopped producing butyric acid because of product inhibition. Fermentation with concentrated hydrolysate under a repeated-batch mode was then conducted to evaluate the stability of butyric acid production during the fermentation process. Samples were taken at the indicated time points for cell growth and product composition analyses.

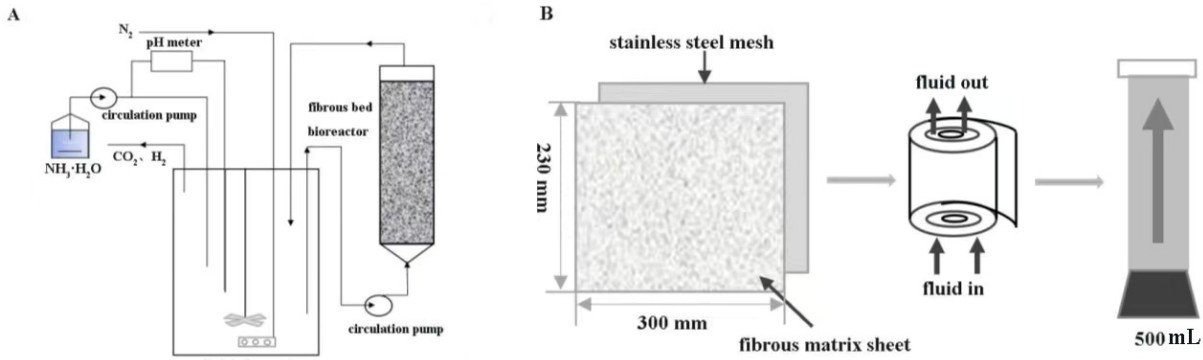

**Figure 1.** Bioreactor system in the process study. (**A**) shows schematic diagram of fibrous bed immobilization reactor; (**B**) shows schematic diagram of fibrous bed assembly.

### 2.7. Analytical Methods

The cell density of the fermentation broth was determined at a wavelength of 600 nm ($OD_{600}$) with a spectrophotometer (Thermo Fisher Scientific Co., Ltd., Shanghai, China).

The high-performance liquid chromatography (HPLC) system (Agilent Technologies, Co., Ltd., Santa Clara, CA, USA) was used to analyze the glucose and xylose contents. The HPLC system consisted of a refractive index detector (RI-2414, Waters Co., Ltd., Milford, CT, USA) and a column oven at 60 °C (Aminex HPX-87H). The eluent was 2.5 mM $H_2SO_4$ at a flow rate of 0.6 mL/min.

A gas chromatograph (GC) system (Agilent Technologies, Co., Ltd., Santa Clara, CA, USA) was used to analyze butyric acid and acetic acid. The GC system consisted of a flame ionization detector (FID) and a capillary column (30 m × 0.32 mm × 1.8 μm). The detector temperature and injection temperature were 250 °C and 200 °C, respectively. The carrier gas was nitrogen with a flow rate of 2 mL/min, and the split ratio was 5:1.

## 3. Results and Discussion

### 3.1. Alkali Pretreatment of Rice Husk

Alkali pretreatment is a typical method to break down the solid structure of lignocellulose [18]. To identify the optimal conditions for alkali-treated rice husk, three main process factors (treatment time, alkaline concentration, and temperature) were evaluated for their effects on the conversion of rice husk to TRS (Figure 2). As expected, the content of TRS increased with reaction progress, and it was found to be maximal at 3 h with a 23.6% TRS yield (Figure 2A). Similarly, a positive relationship within certain limits was also observed between the TRS yield and the temperature (Figure 2C). It could be attributed to the degradation of carbohydrates along with increasing reaction time and temperature [19]. However, the TRS conversion was at the peak of 23.53% with the alkaline concentration of 3 wt% and then rapidly decreased (Figure 2B). It was previously shown that a low alkaline concentration might hold cellulose and hemicellulose chains, which is beneficial to conversion [20]. Thus, referring to the above results, we selected the alkaline concentration (3 %), temperature (120 °C), and reaction time (3 h) as the central values for the subsequent orthogonal design.

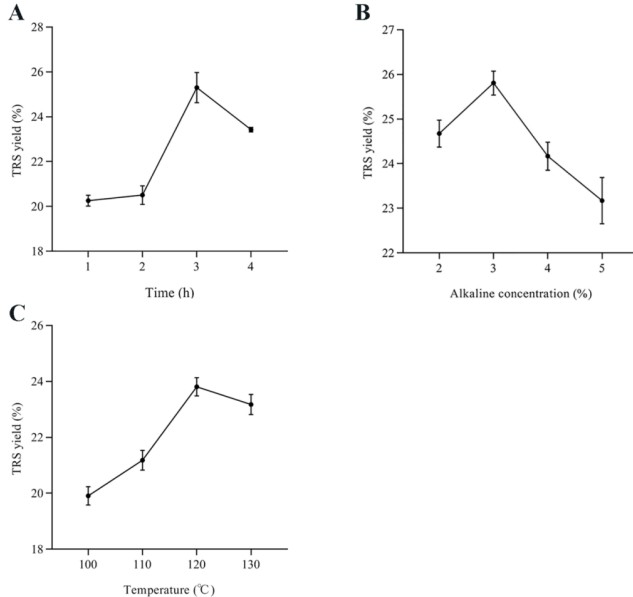

**Figure 2.** Effects of different alkali-pretreated parameters on TRS yield. Panels (**A**–**C**) show the effects of reaction time, alkaline concentration, and temperature on TRS yield, respectively.

An L9 ($3^3$) orthogonal design was used to optimize the above processing conditions. The influences of variables on TRS yield were reflected by *R* values. As shown in Table 3, the temperature showed the greatest impact on the TRS yield, followed by the alkaline concentration and reaction time. Notably, a higher reaction temperature might produce inhibitory compounds and affect TRS production [21]. Based on the *R* value analysis, the optimal condition for TRS yield was selected as follows: treatment time 3 h, alkaline concentration 2 wt%, and temperature 120 °C. Under this processing condition, a TRS yield of 47.34% was obtained from the rice husk hydrolysate.

**Table 3.** Orthogonal analysis of alkali pretreatment parameters.

|  | Factor | | | TRS Yield (%) |
|---|---|---|---|---|
|  | **A** | **B** | **C** |  |
| 1 | 2 | 2 | 110 | 41.21 |
| 2 | 3 | 2 | 120 | 46.74 |
| 3 | 4 | 2 | 130 | 47.34 |
| 4 | 3 | 3 | 110 | 41.26 |
| 5 | 4 | 3 | 120 | 42.07 |
| 6 | 2 | 3 | 130 | 41.69 |
| 7 | 4 | 4 | 110 | 39.73 |
| 8 | 2 | 4 | 120 | 45.06 |
| 9 | 3 | 4 | 130 | 44.54 |
| *K1* | 127.96 | 135.29 | 122.2 | |
| *K2* | 132.54 | 125.02 | 133.87 | |
| *K3* | 129.14 | 129.29 | 133.57 | |
| $k_1$ | 42.65 | 45.1 | 40.73 | |
| $k_2$ | 44.18 | 41.67 | 44.62 | |
| $k_3$ | 43.05 | 43.1 | 44.52 | |
| *R* | 1.53 | 3.43 | 3.89 | |

$k_1$–$k_3$ values are the mean values of TRS yield for each factor at levels 1–3, respectively. A larger *R* value indicates a greater effect of the factor.

## 3.2. Enzymatic Hydrolysis of Rice Husk

Rice husks are rich in lignocellulose by-products, which contain (*w/w*) 37% cellulose, 15% hemicellulose, 20% lignin, and 20% ash [22]. Combining enzymatic hydrolysis with the chemical pretreatment process could enhance the conversion of cellulose to reducing

sugars [23]. Therefore, the alkali-pretreated rice husk was subjected to the enzymatic hydrolysis process. As shown in Figure 3A,C, the TRS yield reached the maximum when the ratio of cellulase to *β*-glucosidase and temperature were 1 and 65 °C, respectively. It is apparent that the TRS production generally increased with reaction time and enzyme amount, but it decreased from the reaction time of 60 h and the enzyme amount of 60 U (Figure 3B,D). Almost no changes in TRS yield were observed between 48 h and 60 h. Therefore, 48 h was set as the central point for the subsequent experiment.

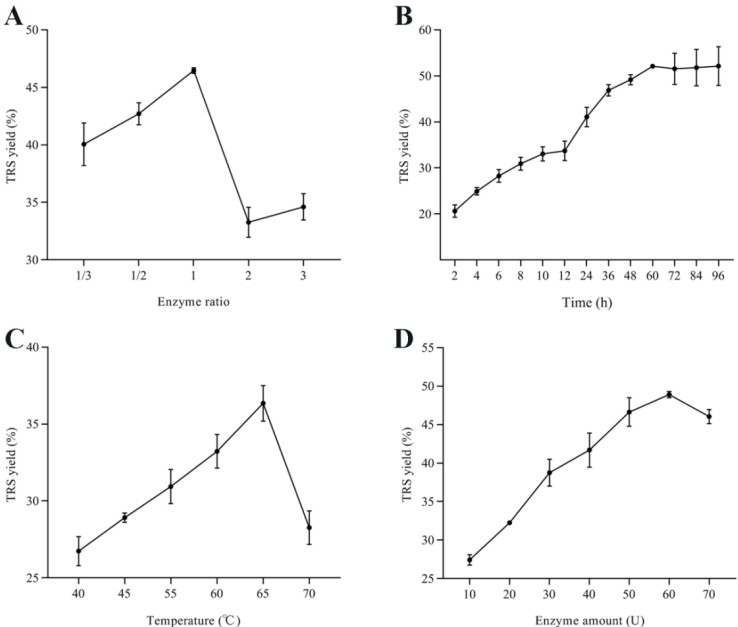

**Figure 3.** Effects of enzymatic hydrolysis parameters on TRS yield. Panels (**A**–**D**) show the effects of enzyme ratio, reaction time, temperature, and enzyme amount on TRS yield, respectively.

The RSM study for reaction parameters, including the enzyme amount, temperature, and reaction time, is shown in Tables 4 and 5.

**Table 4.** Observed responses in the Box–Behnken design for enzymatic hydrolysis.

| | **Independent Variables** | | | **TRS Yield (%)** | |
|---|---|---|---|---|---|
| **Batch** | **A** | **B** | **C** | **Experimental** | **Predicted** |
| 1 | 50 | 50 | 36 | 61.53 | 59.57 |
| 2 | 60 | 60 | 60 | 78.79 | 76.15 |
| 3 | 50 | 50 | 48 | 55.09 | 64.08 |
| 4 | 50 | 70 | 60 | 53.74 | 55.43 |
| 5 | 50 | 60 | 48 | 79.73 | 77 |
| 6 | 40 | 50 | 48 | 55.09 | 54.26 |
| 7 | 50 | 70 | 36 | 57.29 | 55.77 |
| 8 | 50 | 60 | 48 | 76.27 | 77 |
| 9 | 50 | 60 | 48 | 76.05 | 77 |
| 10 | 40 | 60 | 36 | 62.79 | 65.17 |
| 11 | 40 | 70 | 48 | 51.47 | 50.23 |
| 12 | 60 | 50 | 48 | 64.36 | 65.34 |
| 13 | 50 | 60 | 48 | 77.66 | 77 |
| 14 | 40 | 60 | 60 | 68.89 | 68.05 |
| 15 | 50 | 60 | 48 | 75.93 | 77 |
| 16 | 60 | 70 | 48 | 55.59 | 56.16 |
| 17 | 60 | 60 | 36 | 73.50 | 74.08 |

**Table 5.** Analysis of variance table for TRS yield as the response in enzymatic hydrolysis.

| Source | Sum of Squares | df | Mean Square | F-Value | p-Value Prob > F |
|---|---|---|---|---|---|
| Model | 1801.89 | 9 | 168.58 | 31.05 | <0.0001 |
| A | 233.16 | 1 | 144.59 | 26.63 | 0.0004 |
| B | 242.42 | 1 | 87.62 | 16.14 | 0.0003 |
| C | 118.37 | 1 | 12.20 | 2.25 | 0.0027 |
| AB | 25.45 | 1 | 6.64 | 1.22 | 0.0745 |
| AC | 2.93 | 1 | 0.17 | 0.031 | 0.5000 |
| BC | 83.09 | 1 | 7.85 | 1.45 | 0.0069 |
| $A^2$ | 80.49 | 1 | 76.82 | 14.15 | 0.0074 |
| $B^2$ | 977.98 | 1 | 1109.02 | 204.24 | <0.0001 |
| $C^2$ | 14.64 | 1 | 14.64 | 2.70 | 0.9722 |
| Residual | 38.01 | 7 | 5.43 | | |
| Lack of fit | 27.63 | 3 | 9.21 | 3.55 | 0.5341 |
| Pure error | 10.38 | 4 | 2.60 | | |
| Cor total | 1555.23 | 16 | | | |
| $R^2$ | 0.9756 | | | | |
| Adj. $R^2$ | 0.9441 | | | | |
| Predicted $R^2$ | 0.7053 | | | | |

The polynomial equation between the three factors and TRS yield is summarized as follows:

$$\text{TRS yield (\%)} = -726.46018 + 5.55151A + 20.34911B + 2.13156C - 0.012888AB - 0.00170275AC - 0.011676BC - 0.042714A^2 - 0.16229B^2 - 0.012947C^2 \tag{1}$$

where A, B, and C represent the enzyme amount (U), temperature (°C), and reaction time (h), respectively.

The interaction effects of variables for TRS yield are shown using a 3D surface graph (Figure 4A–C). The enzyme amount and temperature and their quadratic ($A^2$ and $B^2$) showed significant effects on TRS yield ($p < 0.01$), which indicated that the enzyme amount and temperature largely affected the TRS yield (Table 5). It should be noted that the temperature had a huge impact on the conversion of reducing sugar, which is in agreement with the fact that the efficiency of enzymatic hydrolysis can be inhibited under nonoptimal temperature conditions due to the protein properties of cellulase [24]. Taken together, the optimal processing conditions for enzymatic hydrolysis were calculated as 23.7 U of cellulase, 23.7 U of β-glucosidase, a hydrolysis temperature of 58.92 °C, and 48 h of reaction time, with a predicted TRS yield of 76.14%. Under these conditions, the enzymatic hydrolysis experiment was repeated three times, and a mean yield of 77.85% was obtained, which meant the model was reliable. A higher yield of TRS is crucial for the following *C. tyrobutyricum* fermentation associated with butyrate production. Therefore, it is suitable to combine an alkali pretreatment with enzymatic hydrolysis for high TRS production.

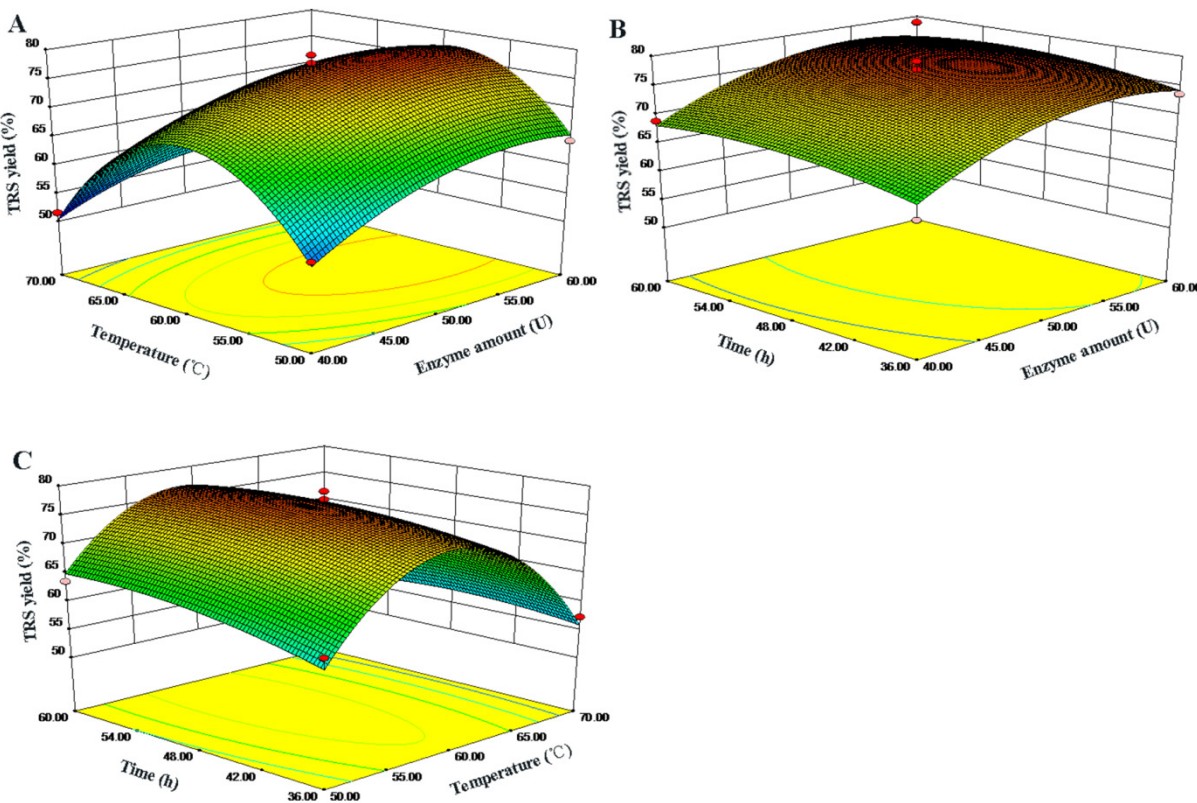

**Figure 4.** The 3D response surface plots showing the effects of temperature (°C), enzyme amount (U), and reaction time (h) on TRS yield. (**A**) shows the interaction between temperature (°C) and enzyme amount (U) on TRS yield; (**B**) shows the interaction between time (h) and enzyme amount (U) on TRS yield; (**C**) shows the interaction between time (h) and temperature (°C) on TRS yield.

*3.3. Butyric Acid Fermentation with Glucose and Rice Husk Hydrolysate in Batch Mode*

To evaluate the possibility of increasing butyric acid productivity, glucose and rice husk hydrolysate were both studied as carbon sources in the free-cell and immobilized-cell fermentations in batch mode (Figure 5). The hydrolysate contained glucose and xylose concentrations with a ratio of 10:1 by HPLC analysis and could be co-utilized in the fermentation. Apparently, butyric acid production and carbon source consumption showed a relatively slow tendency during the first 20 h of fermentation. Because *C. tyrobutyricum* was in an adaptive phase, its cells could not divide rapidly. In batch 1 of the free-cell fermentations, butyric acid and acetic acid were produced cumulatively and reached 17.12 and 4.58 g/L as well as 12.19 and 4.81 g/L when the carbon sources were glucose and hydrolysate, respectively. After adding additional sugars, the final concentrations of butyric acid reached 40.8 and 27.49 g/L, with the corresponding yields of 0.32 and 0.2 g/g and final productivity values of 0.34 g/L/h and 0.21 g/L/h from glucose and hydrolysate, respectively. It should be noted that *C. tyrobutyricum* entered a decline phase in batch 3 due to product inhibition. It could be observed that in batch 2 the yield of acetic acid tended to be stable, reaching 4.9 and 4.96 g/L in the fermentations fed glucose and concentrated hydrolysate, respectively. In addition, the existence of xylose decreased the efficiency of the fermentation of hydrolysate, which may be due to the fact that xylose catabolism usually consumes more energy and is inhibited by glucose [25]. A sharp increase in butyric acid production was observed in the early phase of fermentation with different substrates, which demonstrated that the *pta* deletion inactivated the acetic biosynthesis pathway and shifted the metabolic flux toward butyrate formation [16]. Although there was a small amount of acetic acid production, it had no significant effect on the butyrate increase.

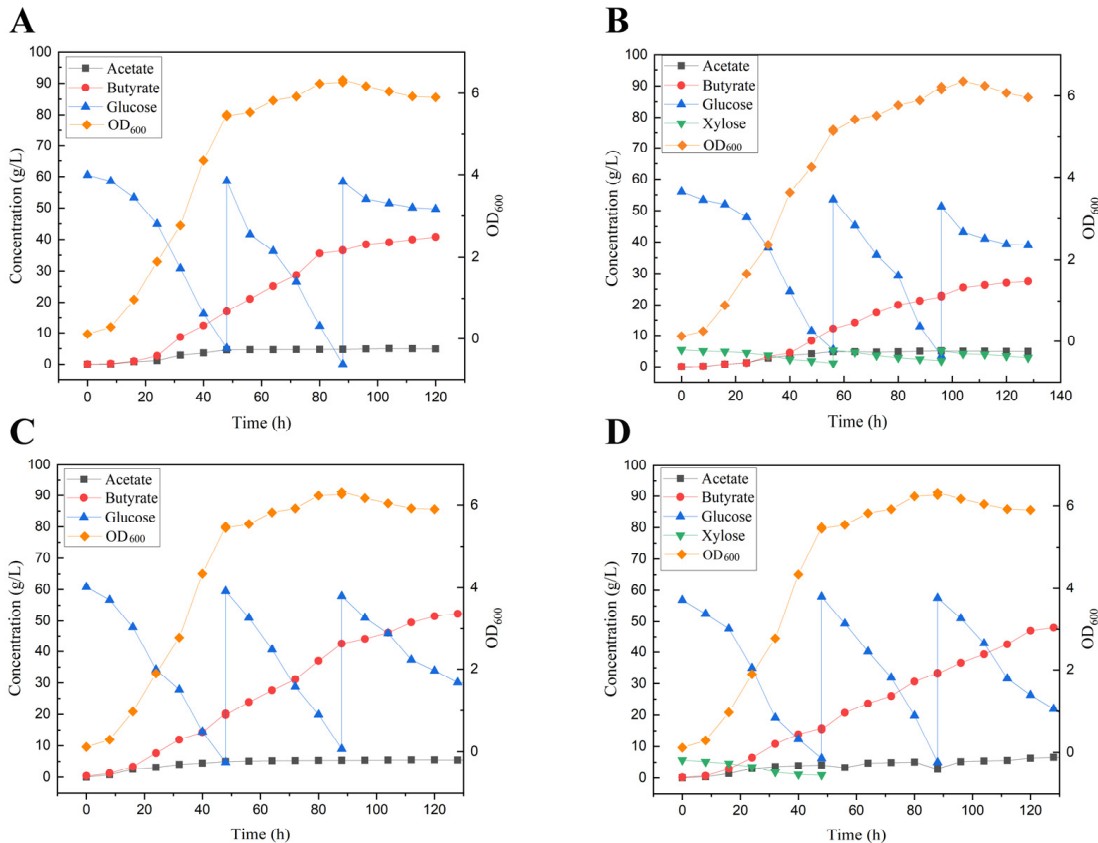

**Figure 5.** Performance of butyric acid production fed with glucose and hydrolysate in batch mode. (**A**,**C**) show the fermentation fed with glucose in the free-cell and immobilized-cell modes, respectively, for butyric acid production. (**B**,**D**) show the fermentation fed with hydrolysate in the free-cell and immobilized-cell modes, respectively, for butyric acid production.

Immobilization cells on support materials can enhance cell loading and product titer and alleviate product inhibition during fermentation [26]. The immobilized *C. tyrobutyricum* was evaluated for its efficiency in utilizing glucose and rice husk hydrolysate, and the results are shown in Figure 5C,D. Compared to free-cell fermentation, the final butyric acid concentration and yield were 53.01 g/L (vs. 40.8 g/L) and 0.36 g/g (vs. 0.32 g/g) from glucose and 49.03 g/L (vs. 27.49 g/L) and 0.3 g/g (vs. 0.2 g/g) from rice husk hydrolysate, respectively, in immobilized-cell mode (Table 6). Apparently, cells immobilized in the FBB showed similar kinetics results with suspended cell fermentation, but much more butyric acid was produced due to the good reactor performance. This is consistent with immobilized cells studies in the FBB [17]. Compared to free-cell fermentation, the butyric acid production from glucose and hydrolysate in the immobilized-cell system increased the butyrate concentration by 29.93% and 78.36%, improved the yield by 12.5% and 50%, and increased the reactor productivity by 14.71% and 71.43%, respectively. The FBB system enhanced the activity of *C. tyrobutyricum* so that during the first 20 h the carbon source consumption was depleted faster than in free-cell mode. It should be noted that the acetic acid yield in FBB mode was higher than that in free-cell fermentation fed with glucose and concentrated hydrolysate because acetic acid yields more ATP to meet the fast metabolism of cell growth [16]. Considering the final product concentration and yield, the immobilized-cell system showed more efficient substrate utilization than free-cell fermentation. However, rice husk hydrolysate as a proper substrate for butyrate production should be studied further. All in all, it is feasible to acquire butyric acid products with rice husk hydrolysate in the FBB.

### 3.4. Butyric Acid Production from Rice Husk Hydrolysate in Repeated-Fed-Batch Mode

To further evaluate hydrolysate for the long-term *C. tyrobutyricum* Δ*pta* fermentation performance, three repeated-fed-batch fermentations of rice husk hydrolysate containing 55.45 g/L glucose and 5.73 g/L xylose were studied in the FBB. As expected, stable butyric acid production was observed in three consecutive batches (Figure 6). It suggested that the immobilized cells were capable of maintaining high metabolic activities without a lag phase, which was consistent with a study showing a similar tendency in repeated-batch fermentation fed with corn husk [27]. The fermentation stopped when the butyric acid concentration reached 16.91 g/L with an average butyrate yield of 0.31 g/g. However, the butyrate concentration and obtained yield were lower than a previous study fed with Jerusalem artichoke (29.7 g/L), probably because of its toxic inhibitors and high ash content (~20%). Moreover, fermentation could be inhibited by the $SiO_2$ in ash [28]. Nevertheless, the immobilized-cell system was feasible for carrying out repeated fermentations, which suggested that the fermentation fed with hydrolysate was stable and reliable.

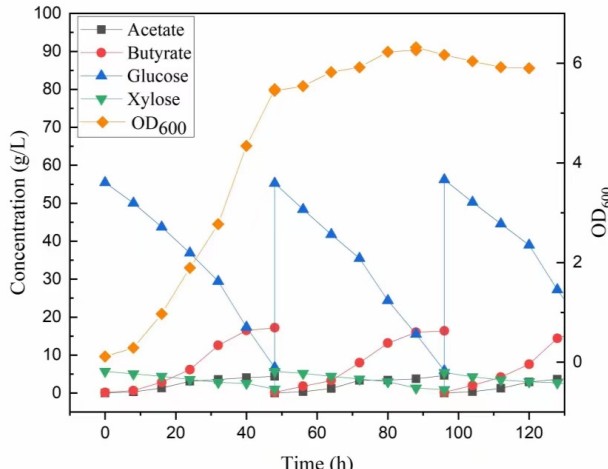

**Figure 6.** Performance of butyric acid production fed with hydrolysate in repeated-fed-batch mode.

### 3.5. Comparison with Other Studies

Table 6 summarizes the fermentation kinetics from various substrates by *C. tyrobutyricum*. The final concentration from rice husk hydrolysate in free-cell mode was obviously higher than that from rice straw (27.49 vs. 18.05 g/L), and a similar fermentation performance was also observed in FBB mode when compared to the results of sugarcane bagasse (49.03 vs. 20.9 g/L). In other words, the final concentrations of *C. tyrobutyricum* in both the free-cell and FBB modes were comparable to those derived from other feedstocks. However, both the yield and productivity in this study were less than that fed with other hydrolysates. This shortcoming could be attributed to the toxic inhibitors of rice husk hydrolysate, such as furfural and 5-hydroxymethylfurfural (5-HMF) [29]. In addition, in FBB, the butyric acid production was simultaneously increased, which could inhibit cell growth. Many methods have been reported to reduce the effects of inhibitors, such as genetic mutation [30], detoxication [31], and immobilization [26]. Therefore, finding ways to improve butyric acid production from rice husk hydrolysate needs to be studied further.

**Table 6.** Comparison of butyric acid production by *C.tyrobutyricum* using different substrates.

| Strain | Substrate | Fermentation Mode | Concentration (g/L) | Yield (g/g) | Reactor Productivity (g/(L × h) | Reference |
|---|---|---|---|---|---|---|
| *C.t* 25755 | SCB | Fed-batch in FBB | 20.9 | 0.48 | 0.51 | [6] |
| *C.t* 25755 | Corn husk | Fed-batch in FBB | 20.8 | 0.39 | 0.42 | [27] |
| *C.t* 25755 | Rice straw | Fed-batch | 18.05 ± 0.03 | 0.36 ± 0.03 | _ | [32] |
| *C.t* 25755 | JA | Repeated-fed-batch in FBB | 27.5 | 0.44 | 2.75 | [33] |
| *C.t* 25755 | SSB | Batch | 11.3 | 0.29 | 0.25 | [34] |
| *C.t* Δ*pta* | Glucose | Fed-batch | 40.8 | 0.32 | 0.34 | This study |
| | RHH | Fed-batch | 27.49 | 0.2 | 0.21 | |
| | Glucose | Fed-batch in FBB | 53.01 | 0.36 | 0.39 | |
| | RHH | Fed-batch in FBB | 49.03 | 0.3 | 0.36 | |

FBB: Fibrous bed bioreactor; SCB: Sugarcane bagasse; JA: Jerusalem artichoke; SSB: Sweet sorghum bagasse; RHH: Rice husk hydrolysate.

## 4. Conclusions

An alkali pretreatment combined with enzymatic hydrolysis was successfully utilized in releasing the TRS of rice husk, and the rice husk hydrolysate could be further used as the carbon source for fermentation by free and immobilized *C. tyrobutyricum* Δ*pta*, producing 27.49 and 49.03 g/L butyric acid, respectively. High and stable levels of butyrate production could be observed with immobilized *C. tyrobutyricum* Δ*pta* in repeated-fed-batch mode. This is the first study to demonstrate the feasibility of butyric acid production from rice husk hydrolysate, and it will widen the thinking of using biomass resources to produce this value-added product.

**Author Contributions:** Y.L.: Conceptualization, Methodology, and Writing—original draft. W.S.: Resources and Execution of the analyses. G.W.: Data curation and Writing—review and editing. H.C.: Data curation and Writing—review and editing. X.P.: Data curation and Writing—review and editing. Y.J.: Data curation and Writing—review and editing. S.-T.Y.: Supervision. M.W.: Project administration, Funding acquisition, Supervision, Writing—review and editing. All authors have read and agreed to the published version of the manuscript.

**Funding:** This work was financially supported by The Science and Technology Key Projects of Zhejiang Province, China (2019C02005), and The National Key Research and Development Program of China (2018YFE0112700).

**Institutional Review Board Statement:** Not applicable.

**Informed Consent Statement:** Not applicable.

**Data Availability Statement:** The data presented in this study are available on request from the corresponding author.

**Conflicts of Interest:** The authors declare no competing interests.

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
