# Peer review of "Production of Butyric Acid from Hydrolysate of Rice Husk Treated by Alkali and Enzymes in Immobilized Fermentation by Clostridium tyrobutyricum CtΔpta"

_fermentation, doi:10.3390/fermentation8100531_

Round 1

Reviewer 1 Report

The study on Production of butyric acid from hydrolysate of rice husk treated by alkali and enzymes in immobilized fermentation by 3 Clostridium tyrobutyricum CtΔpta presents interesting findings. However, following queries should be addressed before considering further processing.

Line #63 and 73: Rephrase the sentence. It is not complete

Line #63: cellulose (Habio Biotechnology…) – should be cellulose

Line #70: mention the type of alkali used

Line #125: concentrated rice husk hydrolysate - elaborate how the rice husk hydrolysate was concentrated

I suggest to move the Fig. 1 and 2 to the Results and discussion part

Mention the software used to do the BBD

Table 4 - Include the predicted TRS yield (%) along with the experimental yield

Include the R2, predicted R2 and the p values for the BBD

Author Response

Response to Reviewer 1 Comments

Point 1: Line #63 and 73: Rephrase the sentence. It is not complete

Reponse 1: We reorganized the sentences of Line 63 and Line 73.

Point 2: Line #65: cellulose (Habio Biotechnology…) – should be cellulose

Response 2: “cellulose” in Line 65 was corrected to be “cellulase”.

Point 3: Line #70: mention the type of alkali used

Response 3: NaOH as the type of alkali was added in Line 70.

Point 4: Line #125: concentrated rice husk hydrolysate - elaborate how the rice husk hydrolysate was concentrated 

Response 4: We added an elaboration regarding how the rice husk hydrolysate was concentrated in line 105. Because the word "concentrated rice husk hydrolysate " first appeared in line 105.

Point 5: I suggest to move the Fig. 1 and 2 to the Results and discussion part

Response 5: We have moved the Fig. 1 and 2 to the Section 3.1. and 3.2..

Point 6: Mention the software used to do the BBD

Response 6: We added the description of the software used to do the BBD in Line 92-93.

Point 7: Table 4 - Include the predicted TRS yield (%) along with the experimental yield. Include the R2, predicted R2 and the p values for the BBD

Response 7: We have made data supplements in Table 4. The polynomial equation in Line 220 was generated by coded factors. So we updated it, which is generated by actual factors now.

Reviewer 2 Report

Dear Authors,

   This study aimed to evaluate the feasibility of butyric acid production by the fermentation of rice husk.

Research is very interesting as well as it has a scientific value. It is worth emphasizing that the presented study characterizes by novelty. The introduction provides a good, generalized background of the topic that quickly gives the reader appreciation of the scientific relevance and timeliness of the research theme.

However, there are flaws of the manuscript that need to be fixed before publication.

Specific comments on the manuscript are as follows:

  • Lines: 26; subsection 2.7: please use one type of font,
  • The findings of this study aren’t sufficiently described in the context of the published literature. The literature sources are presented including approximately 25% that date back to more than 10 years. Please modify it.
  •  Lack of description for used methods:  response surface methodology (RSM), Orthogonal analysis. Please, complete it.
  • Please recheck thoroughly the whole article and improve its grammatical mistakes.
  • Please recheck references according to the journal guidelines.

From my standpoint, this manuscript is appropriate for publication in Journal – Fermentation, after minor revision given the above aspects.  

Author Response

Response to Reviewer 2 Comments

Point 1: Lines: 26; subsection 2.2: please use one type of font

Response 1: We have used one type of font in Lines: 26; subsection 2.2.

Point 2: The findings of this study aren’t sufficiently described in the context of the published literature. 

Response 2: We have added detailed description of results in section 3.3 and 3.5.

Point 3: The literature sources are presented including approximately 25% that date back to more than 10 years. Please modify it.

Response 3: Thanks. We have add some literature published in recent years in the reference, and corresponding revision was done in the text.

Point 4: Lack of description for used methods:  response surface methodology (RSM), Orthogonal analysis. Please, complete it.

Response 4: Thanks for your precise, we have added the description of methods in Line 81 and 92.

Point 5: Please recheck thoroughly the whole article and improve its grammatical mistakes.

Response 5: We have rechecked thoroughly the whole article and improved its grammatical mistakes.

Point 6: Please recheck references according to the journal guidelines

Response 6:  We have rechecked references according to the journal guidelines.